# Computer Simulations of FIFZN (Fuzzy Inverse-Free Zhang Neurodynamics) with Expected Precision Adaptively Satisfied Handling TVQP (Temporally-Variant Quadratic Programming)

Sicheng Zhu
*School of Electronics and Information Technology*
*Sun Yat-sen University.*
Guangzhou, P. R. China
zhusch3@mail2.sysu.edu.cn

Min Yang
*School of Electronics and Information Technology*
*Sun Yat-sen University.*
Guangzhou, P. R. China
yangmin1221@foxmail.com

Haifeng Hu
*School of Electronics and Information Technology*
*Sun Yat-sen University.*
Guangzhou, P. R. China
huhaif@mail.sysu.edu.cn

Yunong Zhang
*School of Computer Science and Engineering*
*Sun Yat-sen University.*
Guangzhou, P. R. China
zhynong@mail.sysu.edu.cn

Ning Tan
*School of Computer Science and Engineering*
*Sun Yat-sen University.*
Guangzhou, P. R. China
tann5@mail.sysu.edu.cn

*Abstract*—**Some models have been used to solve temporally-variant (as well as time-dependent) quadratic programming (TVQP) in the last few years. But most of the models contain inverse matrix and time-invariant sampling gap, which will leads to computation errors and consumptions that we cannot control and do not want. Recently, a continuous inverse-free Zhang neurodynamics (CIFZN) model, also known as zeroing neural network model, is developed for solving TVQP. A fuzzy system is designed to adaptively adjust the sampling gap under the expected precision. Aided with the fuzzy system, an inverse free discrete model is further proposed and termed the fuzzy discrete Zhang Neurodynamics (FDZN) model. In this paper, verification of the efficacy and superiority of the FDZN model and numerical experiments about four specific TVQP examples are conducted. Most importantly, simulative is performed to illustrate the applicability of the FDZN model.**

*Index Terms*—**Zhang neurodynamics (ZN), temporally-variant quadratic programming (TVQP), fuzzy system, adaptive sampling gap, expected precision.**

## I. Introduction

Temporally-variant problems are more common in scientific research than time-invariant problems, which have attracted much attention of researchers [1]–[5]. As a special type from neural networks [6]–[9], Zhang neurodynamics (ZN) [10]–[12] plays an important role in time-dependent problems solving. In recent years, lots of continuous models and some effective discrete models have been designed and analyzed for solving Temporally-Variant Quadratic Programming (TVQP) [13]–[16]. Continuous models could be divided into two categories: inverse-free [13], [14] and inverse-need [15], [16]. Those two categories both have good performance for solving TVQP in continuous form. But both of them leads to some unexpected problems. Therefore, developing a model that meets our needs is important. Fuzzy control [17]–[20] is a typical intelligent control method to get an adaptive strategy. As for temporally-variant problems solving, the fuzzy control also shows its unique advantages [21]–[25]. For solving TVQP, an adaptive fuzzy control strategy was designed and applied to ZN [23]. In addition, some ZN models with constant/fuzzy parameters are developed for solving TVQP [24]. Inspired by the previous work [21]–[25], the fuzzy control method is utilized in this paper to develop the adaptively satisfied (as well as adaptive sampling gap). Verification of a model is absolutely necessary where it is proposed. So in most of the pages of this paper, simulative experiments about robot control are performed to illustrate the applicability of the FDZNN model.

Based on the above discussions, the idea and organizational structure of this paper are delineated as follows. Section II introduces the TVQP formulation and presents a continuous inverse-free ZN (CIFZN) model, which is constructed by the ZN method in a dual application, laying the groundwork for the subsequent development of discrete models, applying the Euler discretization technique to the CIFZN model, resulting

in the formulation of an Inverse-Free Discrete Zeroth-Order Neural (DZN) model. Moving forward to Section III, a fuzzy system is designed in detail to develop an adaptive satisfied. Leveraging this fuzzy system, an advanced FIFDZN model is proposed. Section IV is dedicated to experimental validations, where four TVQP case studies, each with unique requirements, are examined to demonstrate the effectiveness and advantages of the FIFDZN model over the DZN model. The paper concludes with a summary in the final section, highlighting the principal contributions in a clear and concise manner for enhanced comprehension. For a better understanding, the main contributions are listed as below.

1) The inverse-free DZN model is displayed for solving TVQP for the first time on the basis of the CIFZN model and Euler discretization formula, which is more convenient for computer processing, and program coding.
2) The fuzzy system with two inputs and one output is able to generate the adaptively satisfied under the expected precision. Aided with the fuzzy system, the FIFDZN model is further proposed for solving TVQP, which is more flexible, stable, and intelligent compared with existing results.
3) Computer Simulations of FIFZN is conducted in this paper to validate the effectiveness, superiority, and applicability of the FIFDZN for solving TVQP.

## II. PROBLEM FORMULATION AND ZN MODELS

The problem formulation about TVQP is introduced in this section. Then, the inverse-free CZN model and DZN model are developed for solving TVQP.

### A. TVQP Formulation and Solution Model

TVQP [15], [16], [26] is formulated as

$$\min_{\mathbf{x}(t)} \left\{ \frac{1}{2}\mathbf{x}^{\mathrm{T}}(t)A(t)\mathbf{x}(t) + \mathbf{b}^{\mathrm{T}}(t)\mathbf{x}(t) \right\}, \qquad (1a)$$

$$\text{s.t.} \quad C(t)\mathbf{x}(t) = \mathbf{d}(t), \qquad (1b)$$

where coefficient matrices $A(t) \in \mathbb{R}^{n \times n}$ (being positive definite and symmetric) and $C(t) \in \mathbb{R}^{m \times n}$ (being of full row rank) are smoothly temporally-variant; coefficient vectors $\mathbf{b}(t) \in \mathbb{R}^n$ and $\mathbf{d}(t) \in \mathbb{R}^m$ are smoothly temporally-variant; vector $\mathbf{x}(t) \in \mathbb{R}^n$ is the desired solution to be computed in real time.

By mathematical calculation, the inverse-free CZN model is developed as

$$\dot{\mathbf{z}}(t) = U(t)\big(-\dot{P}(t)\mathbf{z}(t) + \dot{\mathbf{q}}(t) - \lambda(P(t)\mathbf{z}(t) - \mathbf{q}(t))\big), \quad (2a)$$

$$\dot{U}(t) = -U(t)\dot{P}(t)U(t) - \lambda(U(t)P(t)U(t) - U(t)). \quad (2b)$$

The effectiveness of (2) is proved in the following theorem [15], [16], [26]–[28].

*Theorem 1:* With $t \gg 0$, starting from proper initial values $U(0)$ and $\mathbf{z}(0)$, $U(t)$ synthesized by inverse-free CZN model (2) converges to $P^{-1}(t)$, and $\mathbf{z}(t)$ synthesized by inverse-free CZN model (2) converges to the theoretical augmented

solution $\mathbf{z}^*(t)$. That is, the first $n$ elements of $\mathbf{z}^*(t)$ converge to the theoretical solution $\mathbf{x}^*(t)$ of TVQP (1).

### B. Inverse-Free DZN Algorithm

By utilizing Euler discretization formula [15], [16], [26]

$$x_{k+1} = x_k + \tau \dot{x}_k + O(\tau^2) \qquad (3)$$

to discretize (2), the inverse-free DZN model for solving TVQP (1) is developed as

$$\dot{\mathbf{z}}_k \doteq U_k\big(-\dot{P}_k\mathbf{z}_k + \dot{\mathbf{q}}_k - \lambda(P_k\mathbf{z}_k - \mathbf{q}_k)\big), \qquad (4a)$$

$$\dot{U}_k \doteq -U_k\dot{P}_kU_k - \lambda(U_kP_kU_k - U_k), \qquad (4b)$$

$$\mathbf{z}_{k+1} \doteq \mathbf{z}_k + \tau \dot{\mathbf{z}}_k, \qquad (4c)$$

$$U_{k+1} \doteq U_k + \tau \dot{U}_k, \qquad (4d)$$

in which $\doteq$ denotes the computational assignment operator. Meanwhile, the symbol $O(\tau^2)$ is used to denote the error order [29], which means the error is proportional to $\tau^2$. Note that the sampling gap is invariable in inverse-free DZN model (4), i.e., $\tau = t_{k+1} - t_k$ with $k = 0, 1, \cdots$. Besides, $h = \lambda\tau$ is set to an appropriate constant, e.g., $h = 0.2$ is set in this paper. Specifically, the word "inverse-free" may be omitted for simplicity in the following pages in the explicit situation.

## III. INVERSE-FREE FUZZY DZN ALGORITHM

As known, the choice of sampling gap influences the model performance. In this section, a fuzzy system with two inputs (error $e$ and error change $e_c$) and one output $u$ is designed to adaptively adjust the sampling gap. Specifically, $e$ is defined as

$$e = \lg(\varepsilon/e_{\mathrm{p}}),$$

in which $\varepsilon$ is the concerned index (e.g., residual error $||P\mathbf{z} - \mathbf{q}||_2$), and $e_{\mathrm{p}}$ denotes the expected precision. The adaptive sampling gap is designed as

$$\tau_k = \tau_{k-1}/2^u. \qquad (5)$$

Accordingly, ZN design parameter is updated as $\lambda_k = h/\tau_k$. In addition, the value of the sampling gap ranges from $[0.0001, 0.5]$ s in this paper. The following steps [21]–[25] are presented to establish the fuzzy system. In the fuzzy control method, notations NB, NS, ZO, PS, and PB are markers of fuzzy sets [30], which respectively represent negative big, negative small, zero, positive small, and positive big.

*Step 1 (Defining fuzzy sets):* Fuzzy sets of $e$, $e_c$, and $u$ are all defined as $\{\mathrm{NB}, \mathrm{NS}, \mathrm{ZO}, \mathrm{PS}, \mathrm{PB}\}$. The values of inputs and outputs all range from $[-1, 1]$.

*Step 2 (Defining membership functions):* Similar triangle membership functions are defined for $e$, $e_c$, and $u$. In detail,

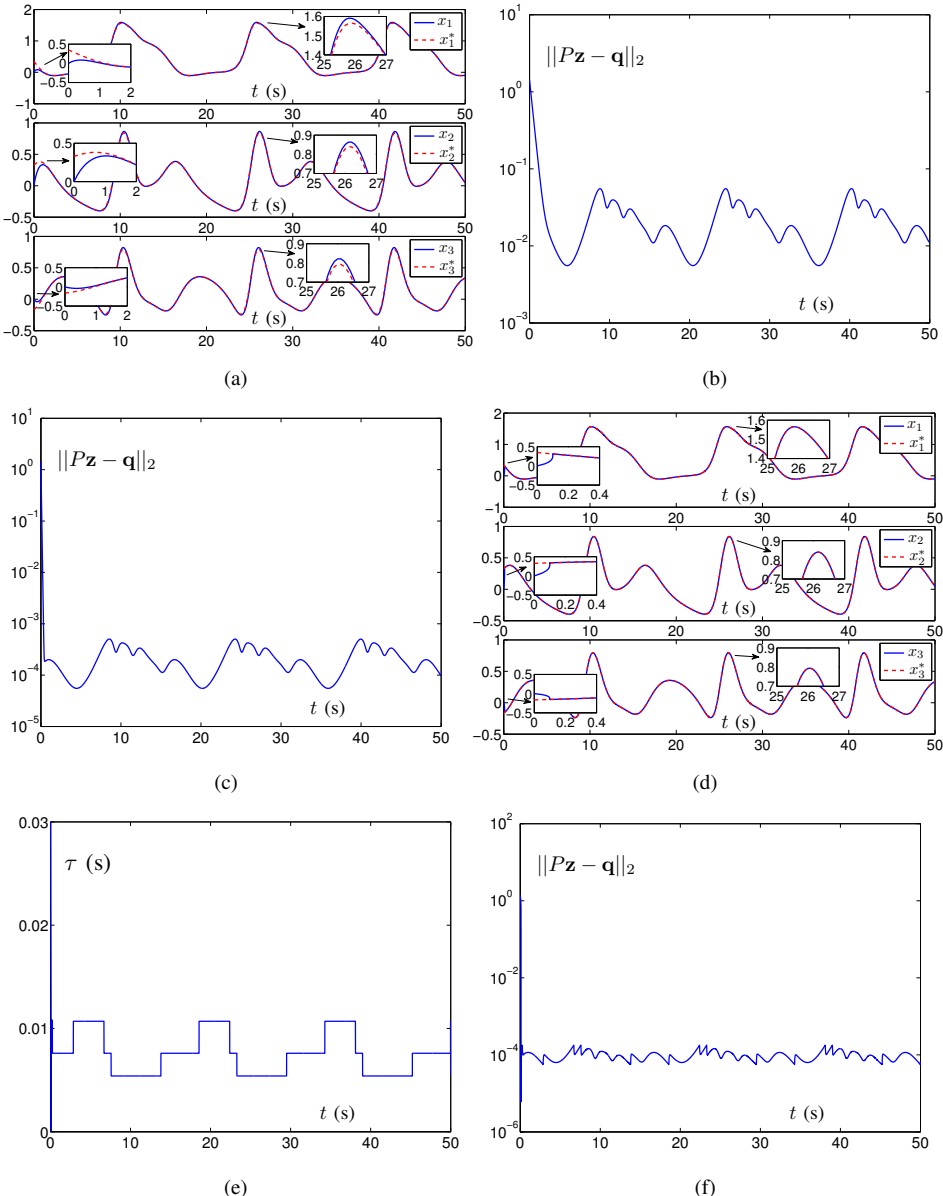

Fig. 1. Results synthesized by inverse-free DZNN algorithm (4) and FDZNN algorithm (6) for solving TVQP in Example 1. (a) Solutions synthesized by DZNN model (4) with $\tau = 0.1$ s. (b) Residual errors synthesized by DZNN model (4) with $\tau = 0.1$ s. (c) Residual errors synthesized by DZNN model (4) with $\tau = 0.01$ s. (d) Solutions synthesized by FDZNN model (6) with expected precision $e_\mathrm{p} = 10^{-4}$. (e) Sampling gaps computed by FDZNN model (6) with expected precision $e_\mathrm{p} = 10^{-4}$. (f) Residual errors synthesized by FDZNN model (6) with expected precision $e_\mathrm{p} = 10^{-4}$.

$\mu_e$ is defined as

$$\mu_e = \begin{cases} \mu_{\mathrm{NB}}(e) = -2e - 1, & -1 \le e \le -0.5, \\ \mu_{\mathrm{NS}}(e) = \begin{cases} 2e + 2, & -1 \le e \le -0.5, \\ -2e, & -0.5 < e \le 0, \end{cases} \\ \mu_{\mathrm{ZO}}(e) = \begin{cases} 2e + 1, & -0.5 \le e \le 0, \\ -2e + 1, & 0 < e \le 0.5, \end{cases} \\ \mu_{\mathrm{PS}}(e) = \begin{cases} 2e, & 0 \le e \le 0.5, \\ -2e + 2, & 0.5 < e \le 1, \end{cases} \\ \mu_{\mathrm{PB}}(e) = 2e - 1, & 0.5 \le e \le 1. \end{cases}$$

*Step 3 (Defining fuzzy rules):* On the basis of the expert experience, 25 fuzzy rules are defined. For instance, *Rule*$_1$*:* if $e$ is NB and $e_\mathrm{c}$ is NB, then $u$ is NB.

*Step 4 (Defuzzifying):* MATLAB provides three types of maximal membership methods for defuzzification [21]–[25]: middle of maximum (MoM), smallest of maximum (SoM), and largest of maximum (LoM). In this paper, mom is used to generate the output $u$ and then update the sampling gap.

The fuzzy system is thus established. Therefore, aided with the adaptive sampling strategy, as well as adaptively satisfied, the inverse-free FDZN model is proposed as

$$\dot{\mathbf{z}}_k \doteq U_k\big(-\dot{P}_k \mathbf{z}_k + \dot{\mathbf{q}}_k - \lambda_k(P_k \mathbf{z}_k - \mathbf{q}_k)\big), \qquad (6a)$$

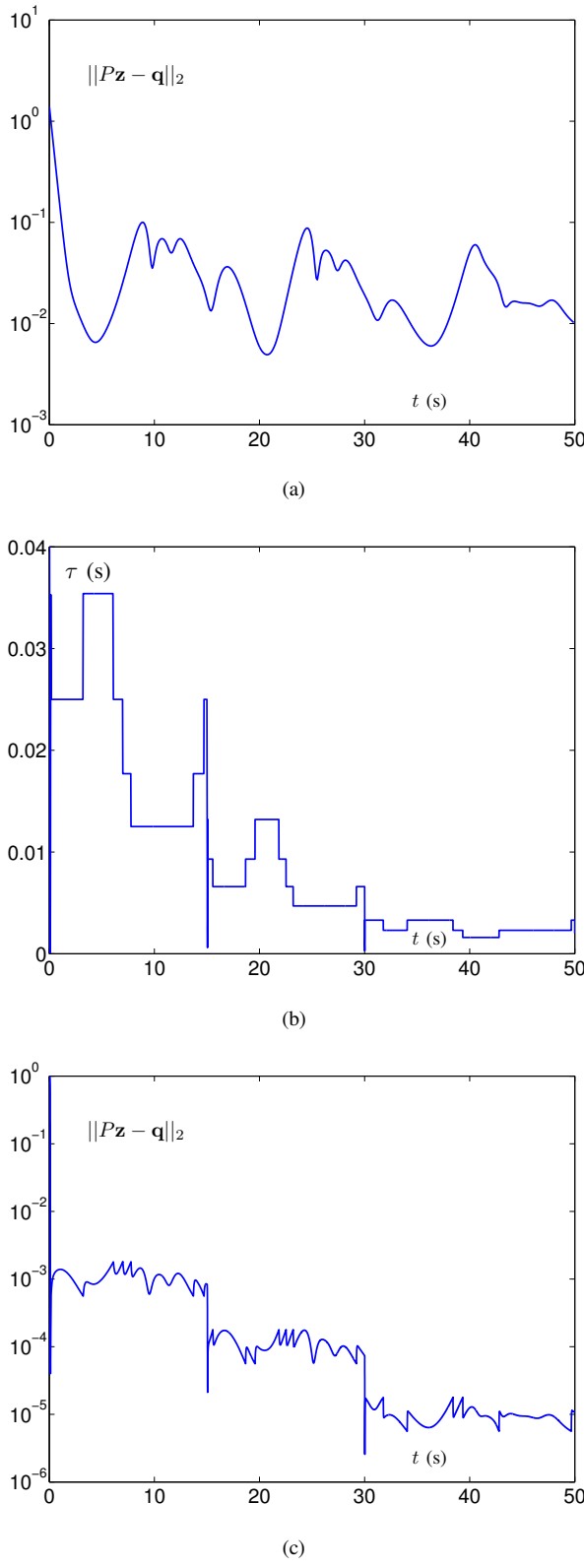

Fig. 2. Results synthesized by inverse-free DZNN algorithm (4) and FDZNN algorithm (6) for solving TVQP in Example 2. (a) Residual errors obtained by DZNN model (4) with $\tau = 0.1$ s. (b) Sampling gaps computed by FDZNN model (6) with expected precisions $e_{\mathrm{p}} = 10^{-3}$ during $[0, 15)$ s, $e_{\mathrm{p}} = 10^{-4}$ during $[15, 30)$ s, and $e_{\mathrm{p}} = 10^{-5}$ during $[30, 50]$ s. (c) Residual errors obtained by FDZNN model (6) with expected precisions $e_{\mathrm{p}} = 10^{-3}$ during $[0, 15)$ s, $e_{\mathrm{p}} = 10^{-4}$ during $[15, 30)$ s, and $e_{\mathrm{p}} = 10^{-5}$ during $[30, 50]$ s.

TABLE I
RESIDUAL ERRORS OBTAINED BY DIFFERENT DISCRETE ALGORITHMS FOR SOLVING TVQP IN EXAMPLE 2

| Residual error | | Time instants (s) | | | | | |
|---|---|---|---|---|---|---|---|
| | | 0 | 10 | 20 | 30 | 40 | 50 |
| Adaptive models | Inverse-free FDZNN | $1.414 \times 10^0$ | $1.016 \times 10^{-3}$ | $9.495 \times 10^{-5}$ | $7.364 \times 10^{-5}$ | $1.251 \times 10^{-5}$ | $1.073 \times 10^{-5}$ |
| | Inverse-need DZNN◊ | $1.414 \times 10^0$ | $9.885 \times 10^{-4}$ | $9.565 \times 10^{-5}$ | $7.417 \times 10^{-5}$ | $1.246 \times 10^{-5}$ | $1.075 \times 10^{-5}$ |
| | DGNN◊ | $1.414 \times 10^0$ | $5.565 \times 10^{-2}$ | $8.671 \times 10^{-3}$ | $7.513 \times 10^{-3}$ | $5.188 \times 10^{-2}$ | $3.504 \times 10^{-3}$ |
| | NI◊ | $1.414 \times 10^0$ | $1.179 \times 10^{-3}$ | $4.670 \times 10^{-5}$ | $4.044 \times 10^{-5}$ | $1.348 \times 10^{-4}$ | $6.192 \times 10^{-5}$ |
| Normal model | Inverse-need DZNN $\tau = 0.1$ s | $1.414 \times 10^0$ | $3.623 \times 10^{-2}$ | $6.541 \times 10^{-3}$ | $2.095 \times 10^{-2}$ | $3.824 \times 10^{-2}$ | $1.054 \times 10^{-2}$ |
| | $\tau = 0.05$ s | $1.414 \times 10^0$ | $1.259 \times 10^{-2}$ | $1.455 \times 10^{-3}$ | $4.706 \times 10^{-3}$ | $1.099 \times 10^{-2}$ | $2.559 \times 10^{-3}$ |
| | $\tau = 0.01$ s | $1.414 \times 10^0$ | $6.434 \times 10^{-4}$ | $5.453 \times 10^{-5}$ | $1.712 \times 10^{-4}$ | $4.810 \times 10^{-4}$ | $9.931 \times 10^{-5}$ |
| | $\tau = 0.005$ s | $1.414 \times 10^0$ | $1.640 \times 10^{-4}$ | $1.354 \times 10^{-5}$ | $4.228 \times 10^{-5}$ | $1.212 \times 10^{-4}$ | $2.474 \times 10^{-5}$ |
| | $\tau = 0.001$ s | $1.414 \times 10^0$ | $6.648 \times 10^{-6}$ | $5.388 \times 10^{-7}$ | $1.675 \times 10^{-6}$ | $4.872 \times 10^{-6}$ | $9.843 \times 10^{-7}$ |
| | DGNN $\tau = 0.1$ s | $1.414 \times 10^0$ | $1.253 \times 10^0$ | $9.008 \times 10^{-1}$ | $1.331 \times 10^0$ | $1.449 \times 10^0$ | $1.016 \times 10^0$ |
| | $\tau = 0.05$ s | $1.414 \times 10^0$ | $1.163 \times 10^0$ | $7.900 \times 10^{-1}$ | $1.291 \times 10^0$ | $1.332 \times 10^0$ | $1.263 \times 10^0$ |
| | $\tau = 0.01$ s | $1.414 \times 10^0$ | $1.076 \times 10^0$ | $8.497 \times 10^{-1}$ | $8.346 \times 10^{-1}$ | $8.662 \times 10^{-1}$ | $6.922 \times 10^{-1}$ |
| | $\tau = 0.005$ s | $1.414 \times 10^0$ | $9.664 \times 10^{-1}$ | $8.396 \times 10^{-1}$ | $7.430 \times 10^{-1}$ | $6.640 \times 10^{-1}$ | $2.608 \times 10^{-1}$ |
| | $\tau = 0.001$ s | $1.414 \times 10^0$ | $5.394 \times 10^{-1}$ | $1.261 \times 10^{-1}$ | $2.179 \times 10^{-1}$ | $3.007 \times 10^{-1}$ | $3.648 \times 10^{-2}$ |
| | NI $\tau = 0.1$ s | $1.414 \times 10^0$ | $2.993 \times 10^{-1}$ | $4.696 \times 10^{-2}$ | $3.934 \times 10^{-2}$ | $1.292 \times 10^{-1}$ | $6.356 \times 10^{-2}$ |
| | $\tau = 0.05$ s | $1.414 \times 10^0$ | $1.485 \times 10^{-1}$ | $2.342 \times 10^{-2}$ | $1.993 \times 10^{-2}$ | $6.598 \times 10^{-2}$ | $3.115 \times 10^{-2}$ |
| | $\tau = 0.01$ s | $1.414 \times 10^0$ | $2.953 \times 10^{-2}$ | $4.673 \times 10^{-3}$ | $4.032 \times 10^{-3}$ | $1.342 \times 10^{-2}$ | $6.199 \times 10^{-3}$ |
| | $\tau = 0.005$ s | $1.414 \times 10^0$ | $1.475 \times 10^{-2}$ | $2.336 \times 10^{-3}$ | $2.019 \times 10^{-3}$ | $6.726 \times 10^{-3}$ | $3.100 \times 10^{-3}$ |
| | $\tau = 0.001$ s | $1.414 \times 10^0$ | $2.949 \times 10^{-3}$ | $4.670 \times 10^{-4}$ | $4.043 \times 10^{-4}$ | $1.347 \times 10^{-3}$ | $6.193 \times 10^{-4}$ |

The symbol ◊ means that the adaptive sampling strategy is combined with the corresponding discrete algorithm.

$$\dot{U}_k \doteq -U_k \dot{P}_k U_k - \lambda_k (U_k P_k U_k - U_k), \qquad (6b)$$

$$\mathbf{z}_{k+1} \doteq \mathbf{z}_k + \tau_k \dot{\mathbf{z}}_k, \qquad (6c)$$

$$U_{k+1} \doteq U_k + \tau_k \dot{U}_k, \qquad (6d)$$

in which $\tau_k = t_{k+1} - t_k$ is temporally-variant, which is updated on the basis of (5) and the fuzzy system.

*Proposition 1:* With $k \gg 0$, starting from proper initial values $U_0$ and $\mathbf{z}_0$, $U_k$ synthesized by inverse-free FDZN model (6) converges to $P_k^{-1}$ with expected precision $e_p$, and $\mathbf{z}_k$ synthesized by inverse-free FDZN model (6) converges to the theoretical augmented solution $\mathbf{z}_k^*$ with expected precision $e_p$. That is, the first $n$ elements of $\mathbf{z}_k^*$ converge to the theoretical solution $\mathbf{x}_k^*$ with expected precision $e_p$.

## IV. EXPERIMENTAL VERIFICATIONS

In this section, experiments about TVQP solving is conducted.

Four examples are considered during time duration $[0, 50]$ s to show the efficacy and superiority of the FDZN model (6) for TVQP solving in different situations. The first example is presented as follows.

*Example 1:* Consider a specific TVQP with coefficient vectors and matrices

$$A(t) = \begin{bmatrix} \sin(\omega t) + 3 & \cos(\omega t) & \cos(\omega t) \\ \cos(\omega t) & \sin(\omega t) + 3 & \sin(\omega t) \\ \sin(\omega t) & \cos(\omega t) & \sin(\omega t) + 3 \end{bmatrix},$$

$$\mathbf{b}(t) = \begin{bmatrix} \sin(\omega t) & \cos(\omega t) & \sin(\omega t) \end{bmatrix}^{\mathrm{T}},$$

$$C(t) = \begin{bmatrix} \cos(\omega t) & \sin(\omega t) + 2 & \sin(\omega t) \\ \sin(\omega t) & \cos(\omega t) & \sin(\omega t) + 2 \end{bmatrix},$$

$$\mathbf{d}(t) = \begin{bmatrix} \cos(\omega t) & \sin(\omega t) \end{bmatrix},$$

in which $\omega = 0.4$.

By setting the initial vector $\mathbf{x}_0 = [0; 0; 0]$, $h = 0.2$, and $\tau = 0.1$ s, solution trajectories synthesized by inverse-free DZN model (4) are presented in Fig. 1(a). The synthesized solutions converge to the theoretical solutions after about 1 s. In addition, the convergence performance is not good enough, e.g., at about 26 s. Specifically, the synthesized residual errors $\|P\mathbf{z} - \mathbf{q}\|_2$ are presented in Fig. 1(b), which are fluctuate between $10^{-1}$ and $10^{-2}$. Meanwhile, if $\tau = 0.01$ s is set for DZN model (4), the synthesized residual errors are fluctuating between $10^{-3}$ and $10^{-4}$, as shown in Fig. 1(c). On the contrary, by setting the expected precision $e_p = 10^{-4}$, solution trajectories synthesized by inverse-free FDZN model

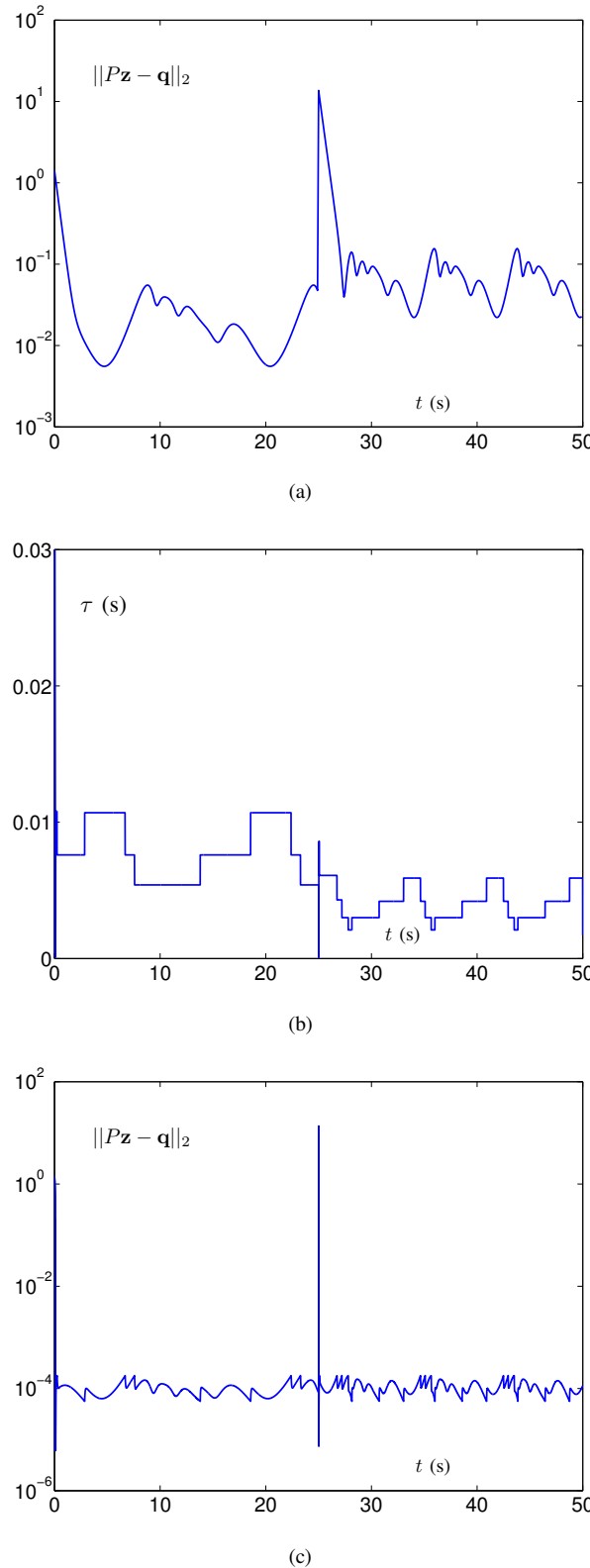

Fig. 3. Results synthesized by inverse-free DZN algorithm (4) and FDZN algorithm (6) for solving TVQP in Example 3. (a) Residual errors obtained by DZN model (4) with $\tau = 0.1$ s. (b) Sampling gaps computed by FDZN model (6) with expected precision $e_{\mathrm{p}} = 10^{-4}$. (c) Residual errors obtained by FDZN model (6) with expected precision $e_{\mathrm{p}} = 10^{-4}$.

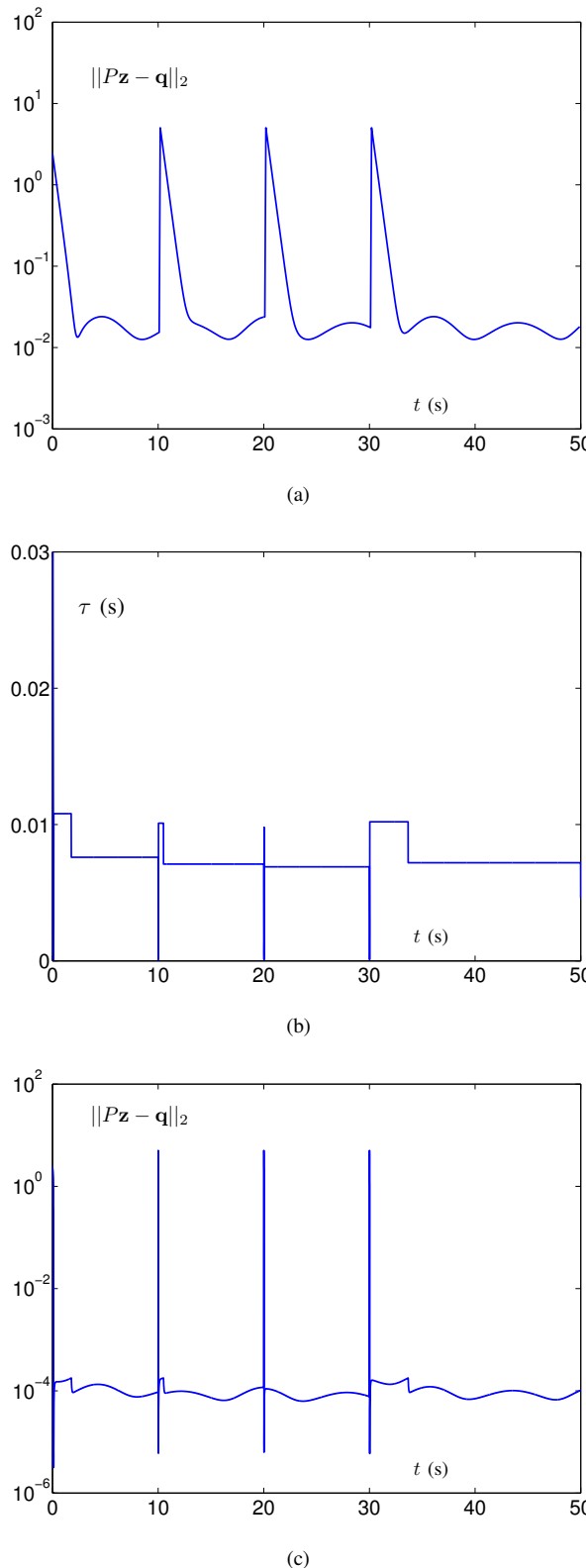

Fig. 4. Results synthesized by inverse-free DZN algorithm (4) and FDZN algorithm (6) for solving TVQP in Example 4. (a) Residual errors obtained by DZN model (4) with $\tau = 0.1$ s. (b) Sampling gaps computed by FDZN model (6) with expected precision $e_p = 10^{-4}$. (c) Residual errors obtained by FDZN model (6) with expected precision $e_p = 10^{-4}$.

(6) are displayed in Fig. 1(d). Evidently, the convergence time is about 0.1 s, and the convergence performance at 26 s is much better. The generated sampling gaps during the solution process are presented in Fig. 1(e), and the precision is indeed consistent with the expected precision as shown in 1(f). The performance of FDZN model (6) is preliminarily substantiated in this example.

*Example 2:* Consider a specific TVQP with coefficient matrix $A(t) =$

$$\exp(\sin(\omega t/4)) \cdot \begin{bmatrix} \sin(\omega t) + 3 & \cos(\omega t) & \cos(\omega t) \\ \cos(\omega t) & \sin(\omega t) + 3 & \sin(\omega t) \\ \sin(\omega t) & \cos(\omega t) & \sin(\omega t) + 3 \end{bmatrix}.$$

Other coefficient information $\mathbf{b}(t)$, $C(t)$, and $\mathbf{d}(t)$ are the same as Example 1 and $\omega = 0.4$. As for the inverse-free FDZN model (6), the precision is desired to be $10^{-3}$ during $[0, 15)$ s, $10^{-4}$ during $[15, 30)$ s, and $10^{-5}$ during $[30, 50]$ s. That is, the precision adjustment of FDZN model (6) is performed in this example.

In this example, two traditional inverse-need models (i.e., inverse-need DZN model and NI model) and an inverse-free model [i.e., discrete GNN (DGNN) model] are developed and compared to substantiate the significance of inverse-free FDZN model (6). Specifically, on the basis of (2), the inverse-need DZN model could be developed as

$$\dot{\mathbf{z}}_k \doteq P_k^{-1} \big( - \dot{P}_k \mathbf{z}_k + \dot{\mathbf{q}}_k - \lambda (P_k \mathbf{z}_k - \mathbf{q}_k) \big), \quad (7a)$$

$$\mathbf{z}_{k+1} \doteq \mathbf{z}_k + \tau \dot{\mathbf{z}}_k. \quad (7b)$$

The NI model [29] could be formulated as

$$\mathbf{z}_{k+1} \doteq P_k^{-1} \mathbf{q}_k. \quad (8)$$

The DGNN model [13] could be designed as

$$\dot{\mathbf{z}}_k \doteq -\gamma P_k^{\mathrm{T}} (P_k \mathbf{z}_k - \mathbf{q}_k), \quad (9a)$$

$$\mathbf{z}_{k+1} \doteq \mathbf{z}_k + \tau \dot{\mathbf{z}}_k. \quad (9b)$$

Not only that, the proposed adaptive sampling strategy (i.e., fuzzy control) is performed in the above three discrete models to show its effectiveness. The corresponding results of inverse-free DZN model (4) and FDZN model (6) are presented in Fig. 2. Evidently, the precision of inverse-free DZN model (4) is determined when $\tau$ is set, and DZNN model (4) does not have the ability of precision adjustment. For example, when $\tau = 0.1$ s, residual errors synthesized by DZN model (4) are presented in Fig. 2(a), which are fluctuate between $10^{-1}$ and $10^{-2}$. On the contrary, FDZN model (6) can adaptively adjust the sampling gap according to the precision requirement, as shown in Fig. 2(b). The expected precision is realized by FDZN model (6) during different intervals, as presented in Fig. 2(c). The detailed comparative results are summarized in Table I. The residual errors at $t = 0$ s, 10 s, 20 s, 30 s, 40 s, and 50 s are listed. Evidently, the adaptive sampling strategy is successfully applied to two ZN models, and the expected precision is realized. However, the expected precision is not realized in DGN model and NI model combined with adaptive sampling strategy. The main reason is that the

two models do not have good convergence performance. In addition, residual errors obtained by those normal models with different sampling gaps are displayed, which fluctuate a lot. The residual errors obtained by the inverse-free DZN model are similar to the inverse-need DZN model and thus are omitted. Although the residual error is relatively small when the sampling gap is small enough, the small sampling gap may result in a huge computation increase. The adaptive sampling strategy guarantees the balance of computation cost and expected precision.

*Example 3:* Consider a specific TVQP with coefficient matrices and vectors being the same as Example 1, but $\omega = 0.4$ during $[0, 25)$ s and $\omega = 0.8$ during $[25, 50]$ s.

By setting $\tau = 0.1$ s, residual errors obtained by inverse-free DZN model (4) are presented in Fig. 3(a), which are fluctuating between $10^{-1}$ and $10^{-2}$ during interval $[0, 25)$ s. At time instant $t = 25$ s, the frequency of TVQP is changed. DZN model (4) returns to the steady state after about 3 s, i.e., at about time instant $t = 28$ s. Not only that, residual errors become larger during interval $[28, 50]$ s because the frequency is changed at $t = 25$ s. On the contrary, inverse-free FDZN model (6) adaptively adjusts the sampling gap to keep the expected precision, as presented in Fig. 3(b). Further, Fig. 3(c) presents residual errors obtained by FDZN model (6). Evidently, at $t = 25$ s, the residual error becomes larger, and a small sampling gap is generated by the fuzzy system. As a result, FDZN model (6) returns to the steady state quickly. After the steady state, the minor adjustment about the sampling gap is completed, and residual errors are always about $10^{-4}$.

*Example 4:* Consider a higher-dimension TVQP with elements of coefficient vectors and matrices being

$$a_{ij}(t) = \begin{cases} \cos(\omega t + i) + 10, & \text{if } i = j, \\ \cos(\omega t + i), & \text{if } i \neq j, \end{cases}$$

$$b_i(t) = \sin(\omega t),$$

$$c_{lj}(t) = \begin{cases} \sin(\omega t + l) + 8, & \text{if } l = j - 2, \\ \sin(\omega t + l), & \text{if } l \neq j, \end{cases}$$

$$d_l(t) = \cos(\omega t),$$

in which $\omega = 0.4$, $i, j = 1, 2, \cdots, 8$, and $l = 1, 2, \cdots, 6$. In addition, three strong interferences are encountered at time instants 10 s, 20 s, and 30 s, respectively.

With $\tau = 0.1$ s, residual errors obtained by inverse-free DZN model (4) are displayed in Fig. 4(a). As shown, DZN model (4) converges to the steady state after about 2 s. After interferences, DZN model (4) returns to the steady state after about 3 s. On the contrary, the convergence speed and restore speed of inverse-free FDZN model (6) are significantly faster than DZN model (4). Specifically, Fig. 4(b) shows the adaptive sampling gap. Fig. 4(c) shows the synthesized residual errors.

## V. CONCLUSION

To establish the foundation for developing discrete models, the CIFZN has been designed for solving TVQP, whose

efficacy has been guaranteed by theoretical analyses. For convenient operation of the digital computer, the inverse-free DZN has been developed by using the Euler discretization formula to discretize the CIFZN. For a better performance, the fuzzy system has been designed to generate the adaptive sampling gap under the expected precision. With the help of the fuzzy system, the FIFDZN with adaptive sampling gap and expected precision has been proposed for solving TVQP. Adequate experiments have been conducted to validate the effectiveness, superiority, and applicability of the FIFDZN for solving TVQP.

### ACKNOWLEDGMENT

This paper is supported by the National Natural Science Foundation of China (with number 62376290). Besides, the corresponding author is Yunong Zhang. In addition, Dr. Min Yang (yangmin1221@hnu.edu.cn) is now with School of Robotics, Hunan University, with the work also supported by the National Natural Science Foundation of China (with number 62303174), the Fundamental Research Funds for the Central Universities (with number 531118010815), and the Changsha Municipal Natural Science Foundation (with number kq2208043).

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
