# OpenReview forum: "Computer Simulations of FIFZN (Fuzzy Inverse-Free Zhang Neurodynamics) with Expected Precision Adaptively Satisfied Handling TVQP (Temporally-Variant Quadratic Programming)"
_IEEE.org/ICIST/2024/Conference — IEEE ICIST 2024 Conference Submission_

### Official Review · Reviewer_cTJb · 2024-08-22
**This article is quite fascinating and of high quality.**

**Rating:** 7
**Confidence:** 3

**Review:**

The paper titled "Computer Simulations of FIFZN (Fuzzy Inverse-Free Zhang Neurodynamics) with Expected Precision Adaptively Satisfied Handling TVQP (Temporally-Variant Quadratic Programming)" utilizes fuzzy systems to propose a novel discrete model, referred to as the Fuzzy Discrete Zhang Neural Dynamics model. Firstly, the inverse-free DZN model is displayed for solving TVQP for the first time on the basis of the CIFZN model and Euler discretization formula, which is more convenient for computer processing, and program coding. Aided with the fuzzy system, the FIFDZN model is further proposed for solving TVQP. Finally, the effectiveness and superiority of the FDZN model are validated through numerical experiments, and simulations demonstrate its applicability. My specific feedback is as follows: 1) In research, what are the key advantages of fuzzy control in solving time-varying problems? 2) Some formatting issues need to be addressed.

---

### Official Review · Reviewer_nhdm · 2024-08-22
**accept**

**Rating:** 7
**Confidence:** 3

**Review:**

This paper develops a continuous inverse-free Zhang neurodynamics model for solving the problem of temporally variant quadratic programming. Then, a fuzzy system is designed to adaptively adjust the sampling gap under the expected precision. Aided with the fuzzy system, an inverse free discrete model is further proposed and termed the fuzzy discrete Zhang Neurodynamics (FDZN) model. Furthermore, simulative is performed to illustrate the applicability of the FDZN model. The following comments should be considered to improve the quality of paper.
1) There are too many abbreviations in the text, and the title also has abbreviations that are inconvenient to read.
2) The contributions should be stressed more in comparison with specific existing works.

---

### Official Review · Reviewer_ynYD · 2024-08-22
**This article is very interesting and a good one**

**Rating:** 7
**Confidence:** 3

**Review:**

In this paper, the inverse-free DZN model was displayed for solving TVQP on the basis of the CIFZN model and Euler discretization formula. The obtained result is valuable and can be accepted if the following problems can be clarified.
(1) In the introduction, the shortages of those relevant studies are suggested to be further summarized.
(2) In the end of Section 1, the organization of this study is suggested to be summarized.
(3) There exist several spelling and grammar errors. Please check carefully and further polish
(4) In the Experiments Verifications, more analysis can be added to better explain the main results of this paper.
(5) The future work is missing in the Conclusion.

---

### Comment · Reviewer_ynYD · 2024-08-21
**This article is very interesting and a good one**

In this paper, the inverse-free DZN model was displayed for solving TVQP on the basis of the CIFZN model and Euler discretization formula. The obtained result is valuable and can be accepted if the following problems can be clarified.
(1)	In the introduction, the shortages of those relevant studies are suggested to be further summarized.
(2)	In the end of Section 1, the organization of this study is suggested to be summarized.
(3)	There exist several spelling and grammar errors. Please check carefully and further polish
(4)	In the Experiments Verifications, more analysis can be added to better explain the main results of this paper.
(5)	The future work is missing in the Conclusion.

---

### Decision · Program_Chairs · 2024-09-06

Accept (Oral)